# Effects of sensorimotor voice training on event-related potentials to pitch-shifted auditory feedback

Sona Patel[1,2]*, Karen Hebert[3], Oleg Korzyukov[1,4], Charles R. Larson[1]

**1** Department of Communication Sciences and Disorders, Northwestern University, Evanston, IL, United States of America, **2** Department of Speech-Language Pathology, Seton Hall University, Nutley, NJ, United States of America, **3** Department of Occupational Therapy, University of South Dakota, Vermillion, SD, United States of America, **4** Department of Communication Sciences and Disorders, University of Wisconsin—Whitewater, Whitewater, WI, United States of America

* Sona.Patel@shu.edu

**Data Availability Statement:** All subject files are available from the Open Science Framework database at: https://osf.io/ef86u/.

**Funding:** This research was supported by a grant from the National Institutes of Health, including

## Abstract

The pitch perturbation technique is a validated technique that has been used for over 30 years to understand how people control their voice. This technique involves altering a person's voice pitch in real-time while they produce a vowel (commonly, a prolonged /a/ sound). Although post-task changes in the voice have been observed in several studies (e.g., a change in mean $f_o$ across the duration of the experiment), the potential for using the pitch perturbation technique as a training tool for voice pitch regulation and/or modification has not been explored. The present study examined changes in event related potentials (ERPs) and voice pitch in three groups of subjects due to altered voice auditory feedback following a brief, four-day training period. Participants in the opposing group were trained to change their voice $f_o$ in the opposite direction of a pitch perturbation stimulus. Participants in the following group were trained to change their voice $f_o$ in the same direction as the pitch perturbation stimulus. Participants in the non-varying group did not voluntarily change their pitch, but instead were asked to hold their voice constant when they heard pitch perturbations. Results showed that all three types of training affected the ERPs and the voice pitch-shift response from pre-training to post-training (i.e., "hold your voice pitch steady" task; an indicator of voice pitch regulation). Across all training tasks, the N1 and P2 components of the ERPs occurred earlier, and the P2 component of the ERPs occurred with larger amplitude post-training. The voice responses also occurred earlier but with a smaller amplitude following training. These results demonstrate that participation in pitch-shifted auditory feedback tasks even for brief periods of time can modulate the automatic tendency to compensate for alterations in voice pitch feedback and has therapeutic potential.

## Introduction

Sensorimotor control is important for achieving accuracy of goal-directed movements and involves active integration of sensory feedback and motor commands during an ongoing

NIDCD R01DC006243 to CL and R03DC013883 to SP (https://www.nidcd.nih.gov/). The funders had no role in study design, data collection and analysis, decision to publish, or preparation of the manuscript.

**Competing interests:** The authors have declared that no competing interests exist.

movement [1]. Speech-motor control relies on integration of auditory feedback information in order to adjust motor commands to correct for deviations from the intended production and produce clear and fluent speech [2]. To examine the sensorimotor integration process for voice and speech, altered auditory feedback can be used [3, 4]. The pitch perturbation technique or "pitch-shift task" is an established method for examining the sensorimotor system for voice control [3, 5–9]. In this technique the auditory feedback is manipulated by changing the voice pitch while a person is speaking, which results in a perceived mismatch between the intended and perceived vocalizations. Deviations in the auditory feedback from the intended voice pitch result in predictable modifications to the voice. These modifications are typically compensatory and in the opposite direction of the pitch shift (an "opposing" response), although voice changes in the same direction of the shift (a "following" response) do occur in both neurologically healthy and impaired individuals [3, 10–12].

Although altered auditory feedback techniques have typically been used to examine sensorimotor control of voice in healthy individuals, research has shown that these techniques can be clinically useful. Delayed auditory feedback is a common therapeutic technique for improving fluency in individuals who stutter and in individuals with Parkinson's disease [13, 14]. Laukkanen [15] demonstrated that by shifting the pitch of auditory feedback while participants read a text aloud, it was possible to change one's habitual pitch. The authors concluded that speaking repeatedly under the influence of auditory feedback changes a person's voice, and as a result, this technique might be useful in voice training and therapy. A number of studies have demonstrated that voice training can affect voice control [16]. For example, vocal training for singing has been shown to affect voice (pitch) control [17–19]. These changes are not only seen at the behavioral level during singing but also at the neurological level [20, 21]. Zarate and Zatorre [21] argue that activities involving activation of sensorimotor and auditory areas are associated with changes in cortical regions as a result of musical practice.

The exact mechanism by which training paradigms that utilize altered auditory feedback modify speech production patterns long-term remains unclear. One possibility is that these paradigms involve sustained attention to the auditory feedback, resulting in improved attentional control during sensorimotor integration and thus speech production. Tumber and colleagues [22] demonstrated the role of attention on vocal compensations to pitch-shift modulations using a dual-task paradigm. Compensatory voice responses were smaller under a dual-task condition in which individuals had to monitor a visual stream of information for target letters while vocalizing during sudden downward pitch shifts of one half of a semitone. This suggests that when less attention was available for the pitch-shift task, individuals were less able to utilize the auditory feedback to change speech production. Additionally, a study by Li and colleagues [23] found that working memory training modified brain activation during a pitch-shift task. In this study participants trained on an adaptive backwards digit span task for 12 days, and brain activity in response to auditory stimuli (event related potentials or ERPs) were measured before and after training during a standard pitch-shift task. Their results showed modifications in the resulting auditory evoked potentials, namely decreased N100 ("N1" component; activity around the 100 ms region of the auditory ERP response) and increased P200 ("P2" component; activity around the 200 ms region of the auditory ERP response) amplitudes recorded during pitch-shift perturbations following training. The working memory training paradigm used in this study involved sustained attention to the auditory domain over multiple training sessions.

Taken together, these studies suggest that training paradigms may improve voice control and related brain processes by causing the individual to engage additional attentional control mechanisms following a sustained focus on auditory attention processing. However, no evidence exists on whether volitional changes to pitch-shifted feedback impact the automatic

error correction processes of voice control. Furthermore, no evidence exists to explain whether simply holding one's voice constant under pitch-shifted auditory feedback passively directs attention to the auditory feedback, thereby impacting voice control and the related neural mechanisms. These factors are important for identifying the minimum parameters under which training paradigms are expected to operate impacting the clinical feasibility of related interventions. Thus, the purpose of this study was to determine whether the neurological mechanisms for voice control are modified due to brief training under pitch-shifted auditory feedback. To consistently change one's voice pattern, it is necessary to go through a period of training that should lead to a stage when the behavior becomes automatic. The automatic nature of this or any movement as a function of training is likely to be reflected in the neural mechanisms underlying that movement [21] and can be examined using event related potentials (ERPs) from electroencephalogram (EEG) recordings. In this study the ERPs were recorded in response to shifts in the voice pitch of one's auditory feedback while vocalizing to assess whether the altered auditory feedback resulted in changes to the automatic compensatory response to alterations in voice pitch feedback. The pattern of auditory-evoked ERPs (i.e., the P50-N1-P2 ERP complex) obtained as a result of speaking under altered auditory feedback have been shown to produce a consistent pattern across studies [24–27] and have been reported to reflect the neural processing of voice pitch feedback perturbations during vocalization [28].

In two variants of the vocal training task implemented in the present study, participants volitionally changed their voice fundamental frequency ($f_o$) during the production of a steady vowel sound. In the third variant, participants did not intentionally vary their voice pitch during vowel production, but instead, were instructed to keep their voice pitch constant. These tasks mirrored those implemented by Hain and colleagues [10] in which participants were also asked to oppose the direction of the shift, follow the direction of the shift, or ignore the shift and maintain a steady pitch. However, we implemented a between-groups design where each group performed a single task: oppose the shift (the "opposing group"), follow the shift (the "following group"), and ignore the shift and maintain a steady pitch (the "non-varying group"). The opposing and following tasks were performed by different groups because differences in the vocal response during each task were observed by Hain and colleagues [10] and also because a multiple baseline approach would not be practical (3 tasks would require 6 EEG sessions, resulting in 15 days of testing or 27 hours). In addition, both voice responses and auditory-motor ERPs were measured during a baseline "maintain a steady pitch" pitch-shift task in a pretest-posttest design, specifically, before and after four training sessions. This allowed us to examine the effects of short training intervals (a few sessions) as would be encountered in typical therapy sessions on voice pitch regulation in typical individuals.

The specific aims of this study were to examine the impact of three brief volitional training paradigms on 1) auditory-motor ERPs (the N1-P2 complex) and 2) voice responses in a pitch-shift task. We predict 1) shorter latencies in the N1 and P2 auditory motor response following volitional training; 2) larger amplitudes in the N1 and P2 auditory motor response following volitional training; and 3) shorter latencies and amplitudes in the voice response during a pitch-shift task following training.

## Methods

### Participants

Thirty-eight participants were recruited from Northwestern University. All participants were native speakers of American English and self-reported being right-hand dominant. They all had normal hearing at octave intervals from 250 Hz to 8000 Hz at 20 dB HL [29] and passed

tests of central auditory processing ("CAP"; the Duration Pattern Sequence test and Pitch Pattern Sequence test [30, 31]) with a score of at least 90% (18 of 20) correct. Participants reported having no history of neurological, speech, or language disorders and minimal vocal training (defined as less than three years of vocal training) and that they did not regularly sing in a group (two times per week or less). Participant recruitment and testing procedures were approved by the Northwestern University Institutional Review Board.

All participants were randomly assigned to one of three training groups: opposing, following, or non-varying (described in the next section). Of the participants recruited, one did not complete the study, two were dropped as a result of not being able to perform the training task, two were dropped due to technical errors in data collection, and four were dropped due to artifacts in the EEG signals (mainly due to movements and sleepiness). As a result, a total of 29 participants remained: 10 participants in the opposing group (3 males, 7 females; mean 19.8 years), 9 participants in the following group (3 males, 6 females; mean 21.4 years), and 10 participants in the non-varying group (5 males, 5 females; mean 21.0 years).

## Procedures

All testing took place in a double-walled, sound-treated booth. A visual display was presented on the computer screen instructing the participant to vocalize an /a/ vowel for 5 seconds. A progress bar indicated the length of time to either "Get ready" or "Say aah". Participant vocalizations were recorded using an AKG boomset microphone (model C420; AKG, Vienna, Austria). The voice was amplified with a 10 dB gain using a Mackie mixer (model 1202; Loud Technologies, Woodinville, IL) and presented as real-time feedback using a Sennheiser headset (Sennheiser Electronic Corporation, Old Lyme, CT) during training and Etymotic Research, Inc., (model ER-2) ear inserts (Etymotic Research, Inc., Elk Grove Village, IL) during the baseline task pre- and post-training. During the vocalization the participant's voice pitch was shifted upward or downward by 100 cents (100 cents = 1 semitone) using an Eventide Eclipse Harmonizer (Eventide, Little Ferry, NJ), creating perturbations in the real-time auditory feedback. MIDI software (Max/MSP v. 5.0) was used to present the display and control characteristics of the pitch-shift (direction randomization, timing, and magnitude). The vocalizations, modified voice feedback signal, and control pulses (used as an indicator of the direction of the pitch-shift) were digitized at 10 kHz, low-passed filtered at 5 kHz, and recorded using LabChart Pro software (AD Instruments, Colorado Springs, CO).

To investigate the effects of volitional voice training on a person's involuntary pitch-shift response, a pretest-posttest design was used. Participants underwent a specific task (referred to as the "baseline task") before and after a training period (Days 1 and 5). Vocal training was performed in four sessions, each on a different day within a two-week period (Days 2–5). The total test time for this experiment was 5.5 hours, with no longer than 1.5 hours per session. All participants were monetarily compensated for their participation.

## Training task

During the training task a single 1000-ms long shift in pitch occurred during each vocalization (either 100 cents down or up) with a random onset between 500 ms and 1000 ms after voice onset. Participants were asked to dynamically change their pitch to either volitionally oppose (the "opposing" group) or follow (the "following" group) the direction of the actual shift depending on the group they were assigned to and maintain the new pitch level for the remainder of their breath. Participants in a third group (the "non-varying" group) were simply asked to ignore the changes in their auditory feedback and maintain a constant pitch and loudness level (i.e., hold your voice steady). Thus, the non-varying group did not volitionally change

their voice in response to the stimuli. Participants performed a short practice session of 10 trials before testing. The instructions in the practice session were the same as the main task. Each training session consisted of 4 blocks of 52 vocalizations.

## Baseline task

For the baseline task (Days 1 and 5), all participants were first fitted with a 32-channel Brain Products actiCAP active electrode cap that was connected to the actiCHamp amplifier (Brain Products GmbH, Germany) for EEG recordings. In addition to recording voice samples, event-related potentials were recorded using BrainVision Recorder software (Brain Products, GmbH, Germany) at a sampling rate of 5 kHz and then low-pass filtered at 400 Hz.

Participants vocalized a steady "aah" sound while their pitch was shifted for five, 200-ms segments within each vocalization. The first shift occurred randomly between 700 ms and 1000 ms after vocalization onset, and each successive pitch-shift occurred randomly between 700 and 900 ms after the onset of the previous shift. The "baseline" task used in the pre- and post-testing is commonly used to assess the pitch-shift response, which occurs automatically in response to a brief change in pitch. Because the training task involved volitional modification of voice pitch, a longer time interval was used to reduce the additional memory demands needed to produce the volitional changes in vocalization. In this task participants in all three groups were instructed to ignore changes to their voice and continue to say "aah" at a constant pitch and comfortable level for the length of the progress bar. A total of 52 test vocalizations were recorded before training and after training, which resulted in 260 trials for each measurement (52 vocalizations x 5 pitch shifts per vocalization) with an approximately equal number of upward and downward pitch shifts.

## Data analysis

Since we were interested in the effects of training on the baseline task (the pitch-shift reflex, an indicator of voice control), data analysis was performed for both ERP and voice data during the baseline task on Day 1 (pre-training) and Day 5 (post-training). Data analysis was not performed for the training task, as these results are reported elsewhere [8].

## EEG analysis

The ERPs were obtained by averaging the recorded EEG signals using Brain Products' Analyzer software, synchronized to the onset of the pitch-shift stimulus. Standard preprocessing of the data was performed including filtering (1–50 Hz), segmentation (500 ms segments were selected; 100 ms pre-shift and 400 ms post-shift), artifact rejection (on the frontal channels and those epochs with amplitudes exceeding ±50 µV), normalization of the mean value to 0, and averaging across all trials. In addition, the data were re-referenced to the common reference instead of using the reference electrode, since preliminary results showed high activation at the region of the reference electrode (FCz). This methodology allowed us to make use of the electrode at the FCz location. The N1 and P2 peak amplitudes and latencies were extracted for a subset of the channels showing maximal negative and maximal positive responses, respectively. An automatic search was performed to identify the global minima (N1 peak) and maxima (P2 peak), during the time window of maximal activation.

All statistical analyses were performed in SPSS (v.17; SPSS Inc., Chicago, IL) to compare the ERPs (N1 peak amplitude, N1 peak latency, P2 peak amplitude, P2 peak latency) obtained before and after training (within-subject factor of time: pre-training vs post-training), and whether this difference was affected by the stimulus direction (up, down), electrode (Fz, FCz, Cz), and the between-subjects factor of group (opposing, following, non-varying). For this,

four separate linear mixed models were conducted. Because Mauchly's test indicated that the assumption of sphericity was violated (p < 0.05), Greenhouse-Geisser corrected estimates were used. We were primarily interested in the main effect of time to determine overall changes due to training. Follow-up paired samples t-tests were also performed within each group to confirm differences in time (from pre-training to post-training) for each electrode and stimulus direction.

### Vocal analysis

Voice samples were analyzed in Igor Pro (Wavemetrics, Inc., Lake Oswego, OR), which called upon Praat [32] for $f_o$ detection. Praat was used to develop a wave representing the voice $f_o$ contour, which was used in further analyses. These $f_o$ contours were first segmented into individual trials of 1100 ms duration (400 ms prior to the pitch-shift onset and 700 ms following the pitch-shift onset). Then outliers were removed from each trial using several processes including normalization by setting the mean baseline voice pitch to 0 cents and removal of extreme values (e.g., extraneous background noise) in the vocalization wave prior to the pitch-shift (for *threshold* = 30 cents, where max cents > *threshold*, and min cents < *-threshold* were rejected) and in the entire duration of each trial when vocalization was occurring (for *threshold* = 1000 cents, where the whole wave was rejected if max cents > *threshold* or min cents < *-threshold*). Only responses that opposed the direction of the pitch shift were used. Finally, the trials were averaged within a participant for each condition (+100-cent shifts, -100-cent shifts). The magnitude of the largest upward or downward compensatory peak ("response magnitude") and time that the peak reached maximum amplitude (referred to as the "response latency") was measured for each subject and submitted to statistical testing using general linear mixed models.

Differences in the voice responses were examined between groups for two measures: 1) voice response latency and 2) magnitude of the largest upward or downward compensatory peak. A log-transformation was performed to achieve homogeneity of variance for voice response latency values but was not needed for peak magnitude responses. Linear mixed models were used to test differences in voice response latency and the absolute values of the peak magnitude with the between-subjects factor of training group (opposing, following, non-varying) and within-subjects factors of time (pre-training, post-training) and direction (up, down). Since direction was not a significant factor ($F(1,71) = 0.890$, $p = 0.349$) for voice amplitude, the up and down responses were aggregated, resulting in fixed factors of group (opposing, following, non-varying) and time (pre-training, post-training).

## Results

### ERP results

The ERPs showed that the maximal negative response occurred between 130–160 ms pre-training and 115–135 ms post-training at the following frontal-central electrodes: Cz, FCz, Fz (shown in Fig 1). The maximal positive response occurred between 210–250 ms pre-training and post-training at the same three electrodes (shown in Fig 2). Each subject's N1 and P2 peak information was extracted for these three channels using the above time windows. The grand averaged ERPs for each group are shown for three electrode sites (Cz, FCz, and Fz) during the pre-training and post-training phases in Fig 3.

Linear mixed models for N1 peak latency and N1 peak amplitude show a significant effect of time on N1 peak latency, ($F(1,30) = 32.002$, $p < 0.05$), but not N1 amplitude, ($F(1,30) = 0.179$, $p = 0.675$). The main effect of group was not significant for the N1 peak latency ($F(2,30) = 0.235$, $p = 0.79$) or the N1 peak amplitude ($F(2,30) = 1.04$, $p = 0.36$). Additionally, none of

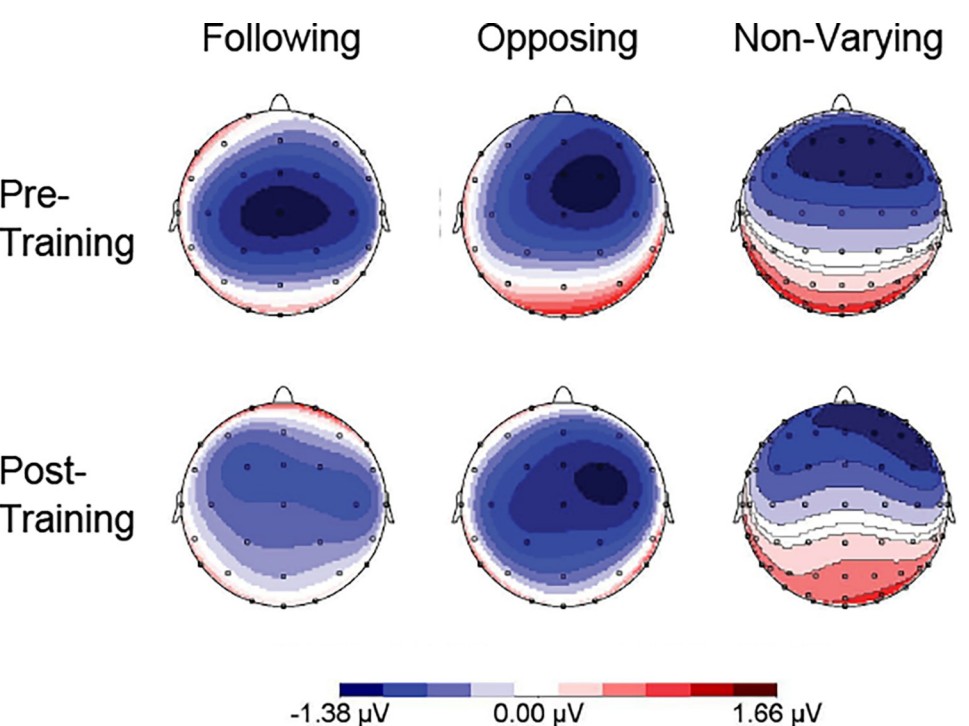

**Fig 1. Mapping view of the N1 response.** Mapping view of the grand averaged ERPs from 130–160 ms pre-training (top row) and 115–135 ms post-training (bottom row) for the following, opposing, and non-varying groups.

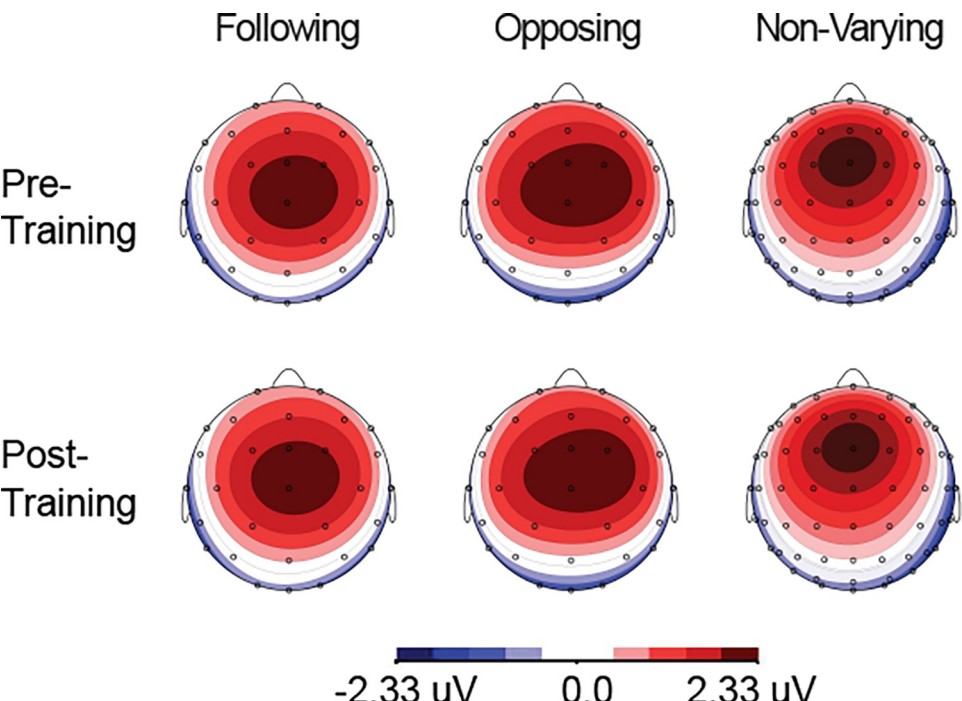

**Fig 2. Mapping view of the P2 response.** Mapping view of the grand averaged ERPs from 210–250 ms pre-training (top row) and 210–250 ms post-training (bottom row) for the opposing, following, and non-varying groups.

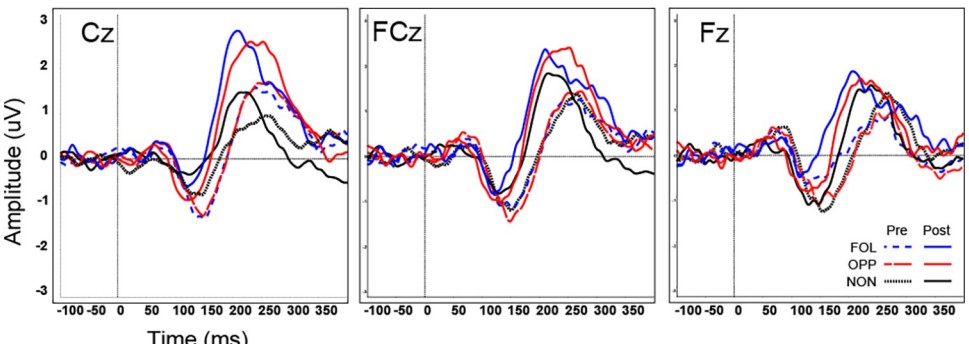

**Fig 3. Grand-averaged ERPs by group at pretest and post-test.** Grand averages of the ERPs at three electrode sites (Cz, FCz, and Fz) for all three groups: opposing (pre: red dashed line; post: red solid line), following (pre: blue dashed line; post: blue solid line), and non-varying (pre: black dotted line; post: black solid line).

the interaction terms for N1 peak latency: group x time ($F(2,30) = 1.685$, p = 0.20), group x direction ($F(2,30) = 1.604$, p = 0.097), group x electrode ($F(4,60) = 2.341$, p = 0.079), time x direction ($F(1,30) = 0.016$, p = 0.902), or time x electrode ($F(2,60) = 0.169$, p = 0.805) or amplitude (group x time ($F(2,30) = 0.234$, p = 0.793), group x direction ($F(2,30) = 0.267$, p = 0.768), group x electrode ($F(4,60) = 0.183$, p = 0.946, time x direction ($F(1,30) = 0.022$, p = 0.884), or time x electrode ($F(2,60) = 1.983$, p = 0.147) were significant. Thus, the N1 peak occurred earlier post-training (M: 129.32 ms; SD: 21.2) compared to pre-training (M: 148.94 ms; SD: 20.5) across all groups. The N1 peak latency and N1 peak amplitude are shown (at the Cz electrode) for all three groups in Fig 4.

A second set of tests was performed to examine the effects of the same four factors (time, direction, electrode, group) on P2 peak latency and P2 amplitude. Greenhouse-Geisser corrected comparisons on P2 peak latency show a significant effect of time (pre-training vs post-training; ($F(1,30) = 25.34$, p < 0.05), but not a main effect of group ($F(2,30) = 0.007$, p = 0.9). Specifically, the P2 peak occurred earlier post-training (M: 230.87 ms; SD: 30.4) compared to pre-training (M: 250.05 ms; SD: 28.6). There was no significant main effect of electrode ($F(2,60) = 1.49$, p = 0.23), although the electrode by group interaction was significant ($F(4,60) = 3.114$, p < 0.05). In other words, the amplitude of the response occurred in different, albeit nearby, electrode sites. There were no significant interactions for P2 peak latency for group x time ($F(2,30) = 1.64$, p = 0.21), group x direction ($F(2,30) = 0.007$, p = 0.993), time x direction ($F(1,30) = 1.34$, p = 0.25), or time x electrode ($F(2,60) = 0.098$, p = 0.9). P2 peak amplitude had a main effect of time ($F(1,30) = 25.55$, p < 0.05) with greater amplitude post-training (M: 1.890 V; SD: .8 V) compared to pre-training (M: 1.448 V; SD: .8 V). No main effect of group was found for P2 amplitude ($F(2,30) = 2.12$, p = 0.13). A significant main effect of electrode was found for P2 amplitude ($F(2,60) = 4.04$, p<0.05) with the amplitude highest at the Cz electrode. The electrode by group interaction was significant ($F(4,60) = 4.9$, p < 0.05). There were no significant interactions for P2 peak amplitude for group x time ($F(2,30) = 0.67$, p = 0.52), group x direction ($F(2,30) = 0.056$, p = 0.94), time x direction ($F(1,30) = 3.26$ p = 0.08), or time x electrode ($F(2,60) = 1.14$, p = 0.31). Because the P2 peak amplitude was largest at Cz this electrode location is plotted for all three groups in Fig 5.

Taken together the findings of a significant effect of time and not a time by group interaction demonstrates a consistent change in ERPs following exposure to training. Specifically, the N1 peak latency and P2 peak latency were reduced, and P2 amplitude was increased from pre-training to post-training across all groups.

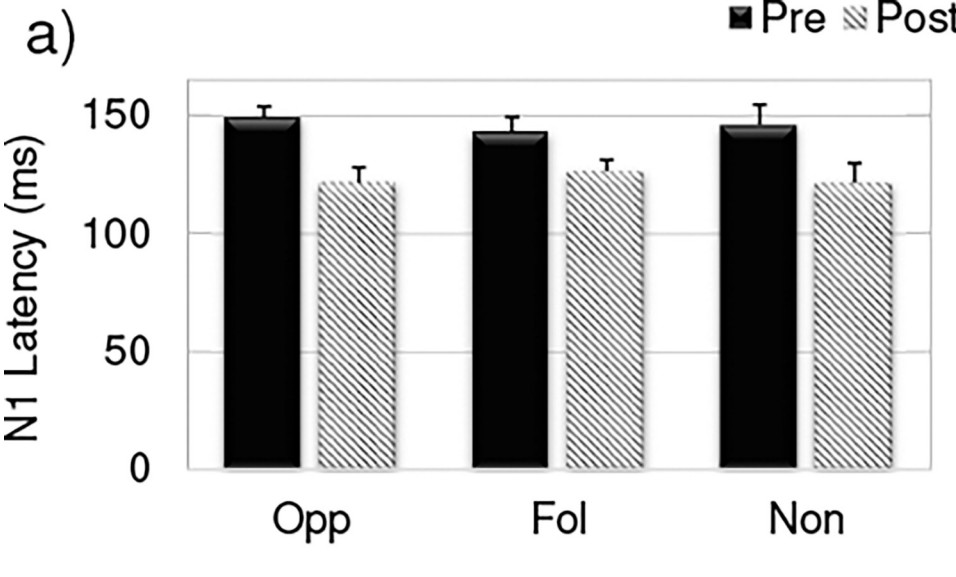

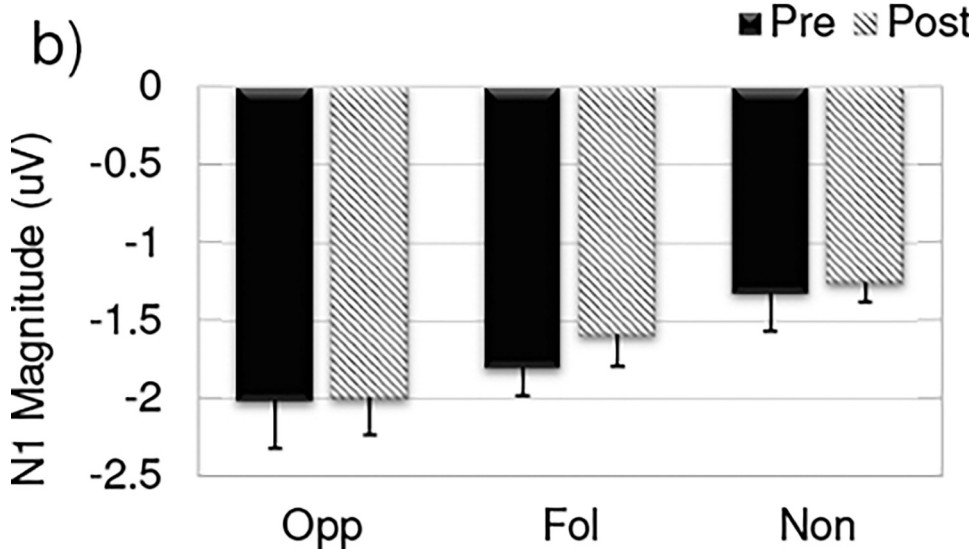

**Fig 4. N1 latency and amplitude.** The a) mean N1 peak latency and b) mean N1 peak amplitude from the ERPs at Cz for the opposing (Opp), following (Fol), and non-varying (Non) groups pre-training (solid bars) compared to post-training (slanted lines). Bars represent standard error of the mean.

### Voice responses

The grand averages of the voice $f_o$ response contours pre-training and post-training are shown for each group in Fig 6 (Panel A: opposing, Panel B: following, and Panel C: non-varying). The voice responses to upward pitch-shifts and downward pitch-shifts are displayed in separate graphs within each panel. The dashed vertical line is the onset time of the pitch-shift stimulus. All groups demonstrate changes in the magnitude of the voice pitch responses from pre-training to post-training.

Linear mixed models showed a main effect of time ($F(1,23) = 7.651$, $p < 0.05$) but not group ($F(2,46) = 2.195$, $p = 0.126$), and no time by group interaction ($F(2,46) = 1.675$,

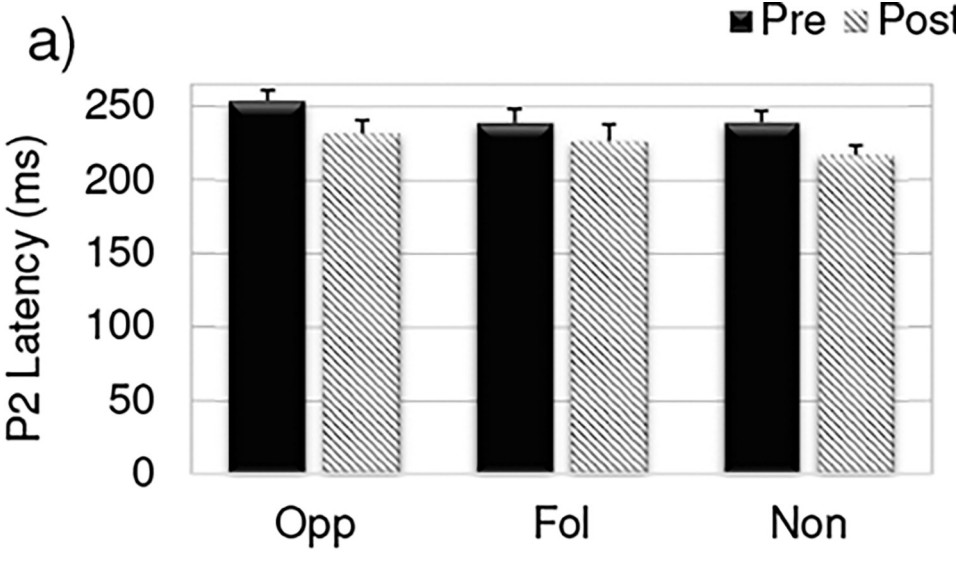

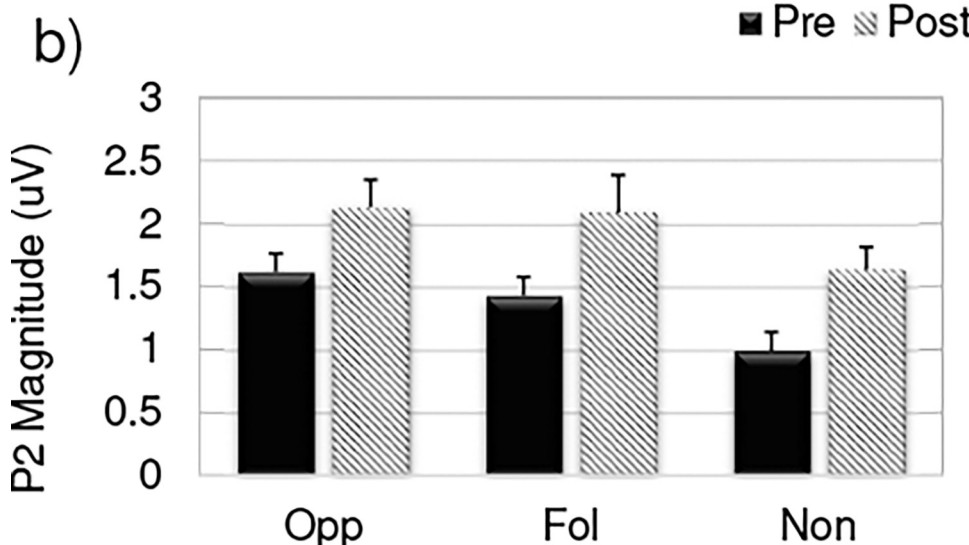

**Fig 5. P2 latency and amplitude.** The a) mean P2 peak latency and b) mean P2 peak amplitude from the ERPs at Cz for the opposing (Opp), following (Fol), and non-varying (Non) groups pre-training (solid bars) compared to post-training (slanted lines). Bars represent standard error of the mean.

p = 0.199) which suggests that response magnitudes for all groups was reduced from the pre-training (M = 24.25, SD = 7.79) to post-training (M = 20.46, SD = 7.02) period similarly. Next, differences in the latency of responses were examined pre- and post-training for each group using a linear mixed model. Results showed main effects of direction ($F_{(1,11)}$ = 5.541, $p < 0.05$) and time ($F_{(1,11)}$ = 5.262, $p < 0.05$) but not group ($F_{(2,22)}$ = 0.249, p = 0.782). The interactions between direction and time ($F_{(1,11)}$ = 3.938, p = 0.073), direction and group ($F_{(2,22)}$ = 3.501, p = 0.071), and time and group ($F_{(2,22)}$ = 1.380, p = 0.273) were not significant. This finding indicates that the latency of the response to the pitch-shift stimulus for all groups was reduced from the pre-training (M = 0.35, SD = 0.12) to post-training period (M = 0.32, SD = 0.17) similarly.

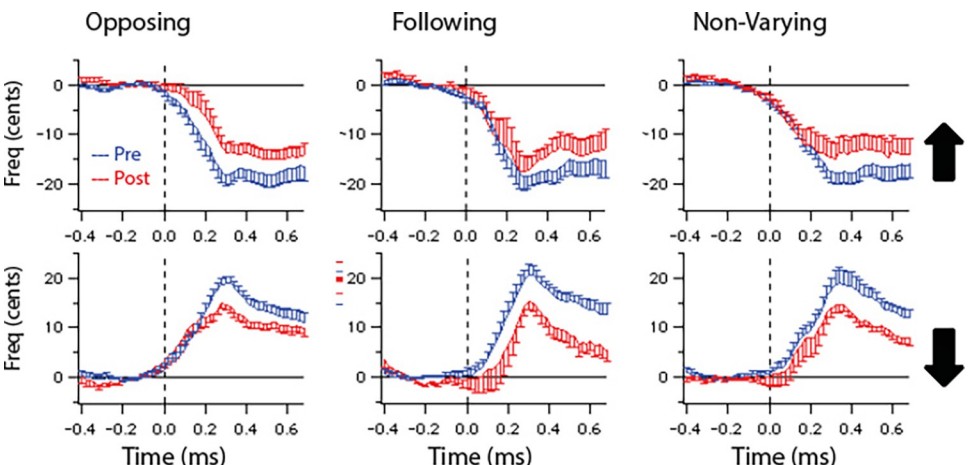

**Fig 6. Grand-averaged vocal responses by group.** Grand-averaged vocal responses to upward (top) and downward (bottom) pitch shifts for the a) opposing, b) following, and c) non-varying groups (blue line represents pre-training responses, red line represents post-training responses).

## Discussion

The purpose of the present study was to determine whether short auditory feedback training intervals could modify voice pitch regulation and affect the corresponding ERPs. Changes in one's auditory feedback while speaking are perceived as errors in production (e.g., [9]). Subjects were trained over a short, four-day period to respond to unpredictable perturbations in the pitch of their auditory feedback as they produced a prolonged vowel sound (/a/). Research has shown that individuals modify their voice in response to errors or simulated changes in their auditory feedback by opposing or following the direction of the change. To better understand the neurological basis of these response types, we trained individuals to attempt to respond in each of these three ways: (a) instructing subjects to change their voice $f_o$ in the opposite direction of pitch-shifted auditory feedback stimuli (opposing the shift), (b) instructing subjects to change their voice $f_o$ in the same direction of the pitch-shifted auditory feedback stimuli (following the shift), or (c) by simply instructing subjects to ignore all pitch shifts and maintain a steady voice pitch (non-varying). A pretest-posttest design was used to assess changes in voice control before and after the training period. The outcome measures included magnitude and latency of the compensatory voice response and the magnitude and timing of the corresponding ERPs to the baseline task, i.e., an involuntary pitch-shift task, where participants were asked to hold a constant pitch and loudness (similar to the non-varying task). The resulting involuntary pitch-shift response is an indicator of voice control.

Our results show differences in the voice responses and the corresponding ERPs during the baseline task as a result of training. Differences in the ERPs were seen in N1 peak latency, P2 peak latency, and P2 peak amplitude. Specifically, both the N1 and P2 peaks occurred earlier post-training compared to pre-training, and the P2 peak magnitude was enhanced post-training compared to pre-training. These results are consistent with the findings of Li and colleagues [23] who report an increase in P2 magnitude post-training. While Li and colleagues [23] found a decrease in the N1 amplitude, we did not find changes to the N1 following training in our study, potentially due to differences in the training task. Other research has shown a N1 suppression (vocalization compared to listening) for pitch-shifts that occur at voice onset cite but a P2 enhancement for pitch shifts that occur mid-vocalization [24, 25]. Behroozmand et al. [24] suggest that this enhancement in the middle of vocalization may reflect an increased

sensitivity or responsiveness to auditory feedback during the resolution of mismatches between the intended vocalization and its feedback. The finding of systematic changes in the neural response suggests that the trained motor behavior (after practice with any of the three instructed conditions of opposing, following, and holding the voice steady) may have become more automatic [21] and the processing of auditory information has become more efficient [23]. These results were complemented by the voice changes, which revealed significant changes in response latency and magnitude in that the peak responses occurred earlier and with a reduced amplitude post-training compared to pre-training. We suspect the reduction in amplitude of the corrective response to pitch-shifts indicates a greater control of the voice, as others have shown larger amplitudes of vocal responses in pathological conditions such as Parkinson's disease, where vocal control is abnormal [33].

The present results confirmed our predictions that ERPs are modified following the two dynamic-response training tasks. Surprisingly, a similar pattern of ERPs was observed for the hold-your-voice-steady ("non-varying") task. Some studies have shown that voluntary responses to the perturbation paradigm engage different mechanisms than involuntary responses in that voluntary responses to pitch shifts can have both involuntary and voluntary components [8, 10, 16]. The involuntary component of responses often results in latency times that are shorter (~100–150 ms) than the voluntary component (~250–600 ms), and are thought to reflect automatic neural processing used by the audio-vocal system to correct for any errors.

On the other hand, the voluntary component is thought to represent higher cognitive mechanisms used at a more conscious level to control voice $f_o$, such as in speaking and singing tasks [8, 10, 16]. In the present study, we investigated involuntary and voluntary responses as individual conditions, rather than as components of the voluntary response. Results show that practicing to maintain a steady pitch also produced differences in ERP/voice responses, potentially because the non-varying task invoked similar cognitive processes used for voice error detection and correction as the other dynamic or volitional tasks. In other words, both dynamic response and hold-your-voice-steady tasks resulted in training an underlying process that positively affected voice pitch regulation. Further, this task activated strong enough processes to produce a change in the neurological mechanism thought to be involved in controlling the voice [5, 23]. Limitations of this study include the relatively small sample per training group. Groups in this study were not balanced by sex although previous work has found that males produce slower and larger vocal responses to pitch shift changes [34]. N1 and P2 amplitude during pitch shift alterations has also been found to vary by gender. The proportions of following vs opposing responses could not be examined and may have changed post-training. Future work should examine the change in response types due to training. This study found that all three training types induced a similar change in vocal response and ERPs to pitch shift stimuli. Future work should also consider examining if training effects vary systematically by gender or age [35].

The results of this study can be used to support the development of brief training interventions for voice modulation. Popular voice therapy programs such as the LSVT® have been shown to be effective in helping individuals with PD to improve their vocal communication by raising voice loudness [36, 37]. However, the standard initial treatment program for LSVT® requires a minimum of 16 sessions over a four-week period. Similar to the LSVT®, the training tasks described in the present study also required subjects to monitor their vocal output and modify their voice $f_o$ based on their auditory feedback. This study found changes in voice control and underlying brain mechanisms supporting speech production in only four brief training sessions. Our findings of behavioral and brain changes due to training suggest that brief voice control paradigms modulate the neurological processes for voice production and

may be valuable in applications for individuals with neurological voice disorders, such as patients with PD.

## Conclusions

In the present study we hypothesized that training individuals to produce a vocal-motor behavior in response to changes in auditory-sensory feedback would affect the ERPs and voice pitch regulation (also assessed as the pitch shift response). Three types of training were implemented where subjects changed their pitch in the opposite direction to a shift in their auditory feedback while vocalizing a prolonged /a/ vowel or in the same direction to the same shift, or subjects maintained a steady voice pitch with no volitional intent to change $f_o$. Effectiveness of training was evaluated by comparing the voice and ERP responses during the baseline task, before and after 4 days of training. Results revealed differences in both the ERPs and voice responses after training for all training tasks. Differences in the ERPs were seen in N1 peak latency, P2 peak latency, and P2 peak amplitude, and voice changes were seen in response latency and magnitude. Changes were seen in ERP responses and voice responses, whereby the peak responses occurred earlier and the peak ERP amplitude was enhanced while the peak voice response amplitude was reduced post-training compared to pre-training. These results suggest that active participation in a vocal task involving the use of altered auditory feedback even for brief periods of time can result in changes in neural activation patterns.

## Acknowledgments

The authors would like to thank Chun Liang Chan for his help with computer programming. The authors have no conflicts of interest to disclose. Information regarding reprints can be directed to Sona Patel, Department of Speech-Language Pathology, 123 Metro Blvd, Nutley, NJ 07110.

## Author Contributions

**Conceptualization:** Charles R. Larson.

**Formal analysis:** Sona Patel, Karen Hebert, Charles R. Larson.

**Funding acquisition:** Sona Patel, Charles R. Larson.

**Investigation:** Sona Patel.

**Project administration:** Sona Patel.

**Software:** Oleg Korzyukov.

**Supervision:** Sona Patel, Oleg Korzyukov, Charles R. Larson.

**Visualization:** Sona Patel.

**Writing – original draft:** Sona Patel.

**Writing – review & editing:** Sona Patel, Karen Hebert.

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
