## [Decision Letter · Decision Letter 0]

4 Aug 2022

PONE-D-22-14364Effects of Sensorimotor Voice Training on Event-Related Potentials to Pitch-Shifted Auditory FeedbackPLOS ONE

Dear Dr. Patel,

Thank you for submitting your manuscript to PLOS ONE. After careful consideration, we feel that it has merit but does not fully meet PLOS ONE’s publication criteria as it currently stands. Therefore, we invite you to submit a revised version of the manuscript that addresses the points raised during the review process.

Two peer reviewers have evaluated your manuscript and have raised a number of queries that need to be carefully addressed in a revision. Please pay particular attention to clarifying the aspects of the study design and methods that the reviewers have identified as needing further explanation.

We look forward to receiving your revised manuscript.

Kind regards,

Jamie Males

Editorial Office

PLOS ONE

Journal Requirements:

Reviewers' comments:

Reviewer's Responses to Questions

**Comments to the Author**

1. Is the manuscript technically sound, and do the data support the conclusions?

Reviewer #1: Partly

Reviewer #2: Yes

2. Has the statistical analysis been performed appropriately and rigorously? 

Reviewer #1: Yes

Reviewer #2: I Don't Know

3. Have the authors made all data underlying the findings in their manuscript fully available?

Reviewer #1: No

Reviewer #2: No

4. Is the manuscript presented in an intelligible fashion and written in standard English?

Reviewer #1: Yes

Reviewer #2: Yes

5. Review Comments to the Author

Reviewer #1: This paper used a pitch-shift paradigm to investigate the training effect on the neuronal and behavioral responses. The experiment had a pre-training baseline and a post-training baseline. In between, four sessions of training took place on four different days, where the participants were instructed differently (to oppose, to follow, or to ignore) to respond to the pitch perturbation. The authors concluded that there was a significant difference between the post-training and the pre-training. The topic is interesting. However, I found that the results of group effect were missing in the report and discussion, while their figures clearly showed a difference among the three groups. Their discussion was not completely consistent with their findings, either. My comments and suggestions can be found below.

Ln 42:

“magnitude of the voice pitch shift response in the baseline” -> Since you have the pre-training and post-training baseline tasks, it would be better to specify them here.

Ln 152:

“vocalize an /a/ vowel” -> prolong for how many seconds?

Ln 171:

I wonder why each session would take up to 1.5 hours. Assume that you asked the participants to vocalize /a/ for 3 or 5 seconds. Each training session had 4 blocks of 52 vocalizations, meaning that they had to say /a/ 208 times. Then, the total time that should be used would be less than an half hour or less than an hour. How long was the inter-vocalization delay? How long was the break between each block?

Ln 182:

“a short practice session of 10 trials before testing” -> What was the instruction for the practice session? Was it the same as the baseline task (to ignore the pitch-shift stimuli)?

Ln 190-191:

Please explain the rationale for using different pitch-shift stimuli in the baseline and the training tasks. In the training tasks, a single 1000-ms long shift per vocalization was used (see Ln 174). However, 5 shifts (each 200 ms-long) per vocalization were used in the baseline tasks. I wonder why.

Ln 220-221:

“We were primarily interested in the main effect of time and the interaction of group x time…” -> It seems that the author(s) preferred a pre-planned comparison. Then, why did the author(s) include “electrode” and “stimulus direction” as the fixed effects in the model? The main effects of electrode and stimulus direction were all missing in the results section. If these factors are not important, I would suggest to remove them from the analysis. If they should be included in the regression model, then I would expect to see the statistics report in the results section (no matter whether they were significant).

Ln 235:

Why was the 1000 cents threshold used for the entire duration, while the 30 cents threshold was used in the pre-shift period?

Ln 236:

“Only responses that opposed the direction of the pitch shift were used.” -> Can you provide the percentage of opposing responses for each group (opposing, following, non-varying)? It would be interesting to see if the participants in the “following” group would have less opposing responses than those in the “opposing” group. I think it is worth discussing whether the training method (i.e., the group effect) played an important role in their responses.

Ln 272 (Fig 4):

Figure 4 shows that there should be a main effect of group for the N1 peak amplitude (as you can see the non-varying group had smaller N1 than the “following” group and the “following” group smaller than the “opposing” group). In your report (from Ln 260 to Ln 272), you did not mention whether the main effect of group was significant or insignificant. As I suggested above, if all the four fixed factors (time, stimulus direction, electrode, group) were included, all the statistics (both significance and insignificance) should be reported here. Currently, many statistics (main effects and interactions) were missing in this paragraph.

For your Figure 4, why did you choose to plot the data for the Cz electrode? What happened to the other electrodes?

Ln 282-283:

“…the electrode by group interaction was significant for both P2 peak latency …and amplitude” -> What are the results of simple main effects analyses? Only significant at the Cz electrode? (Fig5) What happened to the other electrodes?

Ln 285-288:

I feel it’s a bit awkward to say the N1 peak latency effect in this paragraph, as you mainly focused on the statistical results of P2 in this paragraph. A short summary for both N1 and P2 can be placed in a separate paragraph below.

Ln 289:

Your Figures 1, 2, 4 (N1 peak amplitude), 5 (P2 peak amplitude) all showed a main effect of group. However, the statistics were all missing in the text. I think the manipulation of group (i.e., different instructions for the participants) was an important factor in the present study. I would expect a considerable portion of discussion on this issue.

Ln 310:

What was the Levene’s test result of log-transformed data? Please provide the statistics.

Ln 313:

“…changed from the pre- to post-training period …” -> I think the word “change” was a bit vague. It would be better to specify the tendency: larger or smaller. Same for the sentence in Ln 322.

Ln 342:

“The results are consistent with the findings of Li and colleagues [23] who report a decrease in N1 and an increase in P2 magnitude post-training.” -> But you had no significant effect of time in your N1 amplitude. How can you say they are consistent? Or can you explain why the effect of P2 amplitude was present and why the effect of N1 amplitude was absent in your study?

Ln 345-346:

“of the three training tasks” -> What are the three? (opposing, following, non-varying?)

“automatic [21]…efficient [23]” -> Do these two terms refer to peak latency? If yes, then it should be moved to Ln 342, where you discussed your N1 and P2 peak latencies.

Ln 348:

“…with a reduced amplitude post-training compared to pre-training” -> But your results show that P2 peak amplitude was enhanced rather than reduced in the post-training task compared to pre-training. Your discussion here does not match what you found. Please revise.

Ln 363-266:

“Results show that practicing to ignore auditory changes in pitch and hold your voice pitch steady also produced differences in ERP/voice responses, potentially because the non-varying task invoked similar cognitive processes used for voice error detection and correction as the other dynamic or volitional tasks.” -> It seems that the authors try to compare the similarity between pre-training and post-training. However, a more interesting comparison would be whether the instruction during the training session would affect how they respond in the post-training task (i.e., the non-varying task). Please elaborate more on this.

Ln 393-394:

“in both ERPs and voice responses, the peak responses occurred earlier and with a reduced amplitude post-training compared to pre-training.” -> Your N1 was reduced but N2 was enhanced. So this conclusion should be revised, not “both” ERPs.

Reviewer #2: The authors report an investigation of the nature of the pitch-shift reflex before and after a training period to inform whether changes are observed in ERPs and vocal pitch. The study examines an important and relevant question to the field. The motivation and methodological choices for the study require additional information to fully evaluate the contributions. The authors suggest in the introduction and abstract that the current investigation has therapeutic potential, but this notion is not clear from the manuscript in its current form. More information in the motivation and interpretation of the study is required to support this claim throughout.

The introduction requires more detail to help the reader understand the motivation of the current study. In addition, the specific study aims and hypotheses that correspond to the methods are absent from the introduction. This information is required to fully evaluate the appropriateness of the study methods, analysis, and interpretation of the results. Detailed comments below.

Line 59 – 66. It is not clear why understanding the nature of this response would be useful for voice rehabilitation at this point in the manuscript. A discussion of how individuals seeking voice rehabilitation have differing responses to these paradigms is needed.

Line 75 - 76. Please define what the authors intend by ‘voice control’ and ‘pitch control’. If these are referring to the same thing, it would also be helpful to keep terminology consistent as well.

Line 84. Pitch-shift task is not defined. These tasks are variable in the literature, including timing, shift amount, frequency of shifts, duration of shifts, etc. Please include more detail about the paradigms throughout the introduction and review of prior work for clarity.

Line 91. ERP is not defined or explained at this point. The reader needs more information to evaluate what the contribution of this study is to the current study motivation.

Line 92. N1 and P2 are not defined or explained at this point.

Line 95 – 100. The authors motivate the study purpose by saying that no evidence of this yet exists. This could be further strengthened by the addition of why these questions, specifically, need to be answered.

Line 111. This is the first mention of auditory-motor ERPs. Please provide more information here or earlier in the introduction.

Line 116. The authors use inconsistent nomenclature for “f0”. Here, it is f0 and elsewhere in the manuscript the ‘f’ is italicized. It would be helpful to use the notation recently proposed as a consensus; a lower-case italic f and subscript of an ‘o’ for oscillation (see Titze, I. R., Baken, R. J., Bozeman, K. W., Granqvist, S., Henrich, N., Herbst, C. T., ... & Wolfe, J. (2015). Toward a consensus on symbolic notation of harmonics, resonances, and formants in vocalization. The Journal of the Acoustical Society of America, 137(5), 3005-3007.)

Line 116 – 130. It is nicely outlined how the authors are building on the study by Hain and colleagues, however the reason for this additional investigation could be better explained. Specific research aims of the current work are not described. Study hypotheses are also absent. Please include the specific study aims and hypotheses for this work.

Line 147 – 149. The groups are not balanced by sex, which could have impacted the results. Pitch-shift responses have been shown to be impacted by speaker sex (Chen, Z., Liu, P., Jones, J. A., Huang, D., & Liu, H. (2010). Sex-related differences in vocal responses to pitch feedback perturbations during sustained vocalization. The Journal of the Acoustical Society of America, 128(6), EL355-EL360.). This should be acknowledged, along with other study limitations, with references to the specific directional effects that might be expected.

Experimental methods are well described.

For the analysis section, more information and justification are needed for data that were excluded. This information is required to fully evaluate the analysis methods and interpretation of the study. Comments below.

Line 231 – 236. More descriptive statistics are needed for the removed data. E.g., How many trials were removed for each speaker on average, and how many speakers required trials to be removed? A response of greater or less than 10 semitones is very large, how many speakers demonstrated a response outside this threshold? This information informs future work and is important to clarify.

Line 236. Why were only responses that opposed the direction of the pitch shift used? This requires justification. Recent work supports that following responses are common, with one study observing that all of their participants had a proportion of following responses to pitch-shifts (Franken, M. K., Acheson, D. J., McQueen, J. M., Hagoort, P., & Eisner, F. (2018). Opposing and following responses in sensorimotor speech control: Why responses go both ways. Psychonomic Bulletin & Review, 25(4), 1458-1467.). It is not clear why following responses were removed from the current sample when evidence shows it is a consistently observed behavior that is relevant to understanding the nature of the pitch-shift response.

Line 236. It would also be interesting to include if there were differing percentages of following responses by group (opposing, following, and non-varying).

Line 238 – 240. More information is required on the statistical testing. Please include descriptive information about each statistical test (e.g., method, software used, input variables, outcome variables), any corrections that were applied, and the corresponding study aim/purpose for the test.

Line 260. Please include statistical methods and software in the methods section.

Results and discussion – The results appear to be thoroughly described, but the results and interpretation cannot be fully evaluated given earlier comments regarding study hypothesis and analysis methods.

6. PLOS authors have the option to publish the peer review history of their article (what does this mean?). If published, this will include your full peer review and any attached files.

Reviewer #1: **Yes: **Li-Hsin Ning

Reviewer #2: No

---

## [Author Response · Author response to Decision Letter 0]

19 Sep 2022

We appreciate the editor’s and reviewers’ time in providing helpful comments and suggestions which we believe have improved our manuscript (PONE-D-22-14364). We have carefully considered each of the comments identified and hopefully have improved the clarity of our manuscript. Below we list our responses to the suggestions and criticisms and discuss how we have modified our manuscript based on the reviewer feedback.

Reviewer 1:

1. Ln 42: “magnitude of the voice pitch shift response in the baseline” -> Since you have the pre-training and post-training baseline tasks, it would be better to specify them here.

 RESPONSE: We now clarify the wording as below: 

Results showed that all three types of training affected the ERPs (N1 peak latency, P2 peak latency, and P2 peak amplitude) and the response latency and magnitude of the voice pitch shift response in the pre-training and post-training task (i.e., “hold your voice pitch steady” task; an indicator of voice pitch regulation). 

2. Ln 152: “vocalize an /a/ vowel” -> prolong for how many seconds?

 RESPONSE: We now note that participants vocalized for 5 sec.

3. Ln 171: I wonder why each session would take up to 1.5 hours. Assume that you asked the participants to vocalize /a/ for 3 or 5 seconds. Each training session had 4 blocks of 52 vocalizations, meaning that they had to say /a/ 208 times. Then, the total time that should be used would be less than an half hour or less than an hour. How long was the inter-vocalization delay? How long was the break between each block?

RESPONSE: The pre and post sessions with EEG took up to 1.5 hours (a minimum of 30-45 minutes are needed to put on and calibrate the EEG system, with additional time needed to remove the cap and wash the hair), but the individual training sessions without EEG took around 30 minutes, resulting in a total experiment time of 5.5 hours. The ISI was 700-900ms as outlined in the “Baseline Task”.

4. Ln 182: “a short practice session of 10 trials before testing” -> What was the instruction for the practice session? Was it the same as the baseline task (to ignore the pitch-shift stimuli)?

 RESPONSE: The instruction was the same as presented during the training task as noted in this section. The goal of this practice session was to familiarize the subject with the training task.

5. Ln 190-191: Please explain the rationale for using different pitch-shift stimuli in the baseline and the training tasks. In the training tasks, a single 1000-ms long shift per vocalization was used (see Ln 174). However, 5 shifts (each 200 ms-long) per vocalization were used in the baseline tasks. I wonder why.

RESPONSE: The “baseline” task used in the pre- and post-testing is commonly used to assess the pitch-shift response, which occurs automatically in response to a brief change in pitch. Because the training task involved volitional modification of voice pitch, a longer time interval was used to reduce the additional memory demands needed to produce the volitional changes in vocalization. In addition, one of the aims of the study was to see if the volitional responses produced during the training task generalized to the typical pitch-shift response.

6. Ln 220-221: “We were primarily interested in the main effect of time and the interaction of group x time…” -> It seems that the author(s) preferred a pre-planned comparison. Then, why did the author(s) include “electrode” and “stimulus direction” as the fixed effects in the model? The main effects of electrode and stimulus direction were all missing in the results section. If these factors are not important, I would suggest to remove them from the analysis. If they should be included in the regression model, then I would expect to see the statistics report in the results section (no matter whether they were significant).

RESPONSE: We believe it is necessary to examine these variables. All of the nonsignificant statistics are now included. 

7. Ln 235: Why was the 1000 cents threshold used for the entire duration, while the 30 cents threshold was used in the pre-shift period?

RESPONSE: The trial had vocalization while the pre-shift period was not expected to have vocalization. Therefore, the outlier removal identification threshold needed to be adjusted to the signals expected during each time period. We clarified this in text:

“Then outliers were removed from each trial using several processes including normalization by setting the mean baseline voice pitch to 0 cents and removal of extreme values (e.g., extraneous background noise) in the vocalization wave prior to the pitch-shift (for threshold = 30 cents, where max cents > threshold, and min cents < -threshold were rejected) and in the entire duration of each trial when vocalization was occurring (for threshold = 1000 cents, where the whole wave was rejected if max cents > threshold or min cents < -threshold).”

8. Ln 236: “Only responses that opposed the direction of the pitch shift were used.” -> Can you provide the percentage of opposing responses for each group (opposing, following, non-varying)? It would be interesting to see if the participants in the “following” group would have less opposing responses than those in the “opposing” group. I think it is worth discussing whether the training method (i.e., the group effect) played an important role in their responses.

RESPONSE: This is a great point. The purpose of this paper was to focus on the neurological responses to the volitional training paradigm, which hasn’t been done before. Because work has shown that both types of responses occur (following and opposing; see Franken, M. K., Acheson, D. J., McQueen, J. M., Hagoort, P., & Eisner, F. (2018). Opposing and following responses in sensorimotor speech control: Why responses go both ways. Psychonomic Bulletin & Review, 25(4), 1458-1467), and there is the potential for brain activation based on these differential responses to cancel each other out, we only analyzed responses that opposed the direction of the pitch shift to increase the homogeneity of the anticipated brain activation and focus on the aspect of responding to a pitch shift that has typically been utilized in the clinical training literature. 

9. Ln 272 (Fig 4): Figure 4 shows that there should be a main effect of group for the N1 peak amplitude (as you can see the non-varying group had smaller N1 than the “following” group and the “following” group smaller than the “opposing” group). In your report (from Ln 260 to Ln 272), you did not mention whether the main effect of group was significant or insignificant. As I suggested above, if all the four fixed factors (time, stimulus direction, electrode, group) were included, all the statistics (both significance and insignificance) should be reported here. Currently, many statistics (main effects and interactions) were missing in this paragraph.

RESPONSE: The main effect of group was not significant for any of the ERPs (N1 latency, N1 amplitude, P1 latency, P2 amplitude). These non-significant main effects and interactions are now included in the results section. 

10. For your Figure 4, why did you choose to plot the data for the Cz electrode? What happened to the other electrodes?

RESPONSE: We chose to show an illustration of the data at only the Cz electrode for clarity because it is the site of maximum activity. The N1-P2 complex is traditionally recorded from the midline electrode locations and is frequency found to be the largest and clearest at electrode Cz in speech training paradigms (Tremblay, K., Kraus, N., McGee, T., Ponton, C., & Otis, B. (2001). Central auditory plasticity: changes in the N1-P2 complex after speech-sound training. Ear and hearing, 22(2), 79-90; Tremblay, K. L., Billings, C., & Rohila, N. (2004). Speech evoked cortical potentials: effects of age and stimulus presentation rate. Journal of the American Academy of Audiology, 15(03), 226-237).

11. Ln 282-283: “…the electrode by group interaction was significant for both P2 peak latency …and amplitude” -> What are the results of simple main effects analyses? Only significant at the Cz electrode? (Fig5) What happened to the other electrodes?

RESPONSE: We now report the statistics of the main effect of electrode for the P2 amplitude and lack of main effect of electrode for the P2 latency in the results section. We now clarify that for brevity and clarity, Figure 5 is plotted only at the Cz electrode because it was the site of the maximal response. 

12. Ln 285-288: I feel it’s a bit awkward to say the N1 peak latency effect in this paragraph, as you mainly focused on the statistical results of P2 in this paragraph. A short summary for both N1 and P2 can be placed in a separate paragraph below.

RESPONSE: We now move the final line summarizing the N1 peak latency, P1 peak latency, and P2 peak amplitude into a short summary paragraph below. This paragraph now reads:

“Taken together the findings of a significant effect of time and not a time by group interaction demonstrates a consistent change in ERPs following exposure to training. Specifically, the N1 peak latency, P2 peak latency, and P2 amplitude were modulated from pre-training to post-training across all groups.”

13. Ln 289: Your Figures 1, 2, 4 (N1 peak amplitude), 5 (P2 peak amplitude) all showed a main effect of group. However, the statistics were all missing in the text. I think the manipulation of group (i.e., different instructions for the participants) was an important factor in the present study. I would expect a considerable portion of discussion on this issue.

RESPONSE: The main effect of group was not significant for any of the ERP measures. We now include the nonsignificant main effect of group in the results section. However, the main effect of group was not of theoretical interest in this study. It is the group X time interaction would reflects potential changes in voice and ERPs following different instructions (to oppose, follow, or hold voice neutral). This interaction was also not significant, indicating that the instruction type presented did not result in differential ERP responses following training. Our conclusion regarding the lack of interaction is that all 3 groups experienced training effects to a similar degree regardless of the instruction provided. This finding and conclusion is discussed in the discussion section of the manuscript.

14. Ln 310:What was the Levene’s test result of log-transformed data? Please provide the statistics.

RESPONSE: The statistics section has now been updated and re-organized.

15. Ln 313: “…changed from the pre- to post-training period …” -> I think the word “change” was a bit vague. It would be better to specify the tendency: larger or smaller. Same for the sentence in Ln 322.

RESPONSE: We now specify the changes in latency and amplitude in the voice responses following the training period. Specifically, we specify the reduction in latency and the reduced peak amplitude in the voice response following training.

16. Ln 342: “The results are consistent with the findings of Li and colleagues [23] who report a decrease in N1 and an increase in P2 magnitude post-training.” -> But you had no significant effect of time in your N1 amplitude. How can you say they are consistent? Or can you explain why the effect of P2 amplitude was present and why the effect of N1 amplitude was absent in your study?

RESPONSE: We expanded on the findings and clarify where we are and are not consistent with the ERP findings between the Li et al. study and our study. The modified information is below:

These results are consistent with the findings of Li and colleagues [23] who report an increase in P2 magnitude post-training. While Li and colleagues [23] found a decrease in the N1 amplitude, we did not find changes to the N1 following training in our study, potentially due to differences in the training task. Other research has shown a N1 suppression (vocalization compared to listening) for pitch-shifts that occur at voice onset cite but a P2 enhancement for pitch shifts that occur mid-vocalization [24, 25]. Behroozmand et al. [24] suggest that this enhancement in the middle of vocalization may reflect an increased sensitivity or responsiveness to auditory feedback during the resolution of mismatches between the intended vocalization and its feedback.

17. Ln 345-346: “of the three training tasks” -> What are the three? (opposing, following, non-varying?)

 “automatic [21]…efficient [23]” -> Do these two terms refer to peak latency? If yes, then it should be moved to Ln 342, where you discussed your N1 and P2 peak latencies.

RESPONSE: We now clarify that this trained motor behavior is based on practice with the 3 instructed conditions of opposing, following, and holding the voice steady. The included statement ‘trained motor behavior (after practice with the 3 instructed conditions of opposing, following, and holding the voice steady) may have become more automatic [21] and the processing of auditory information has become more efficient [23]’ is not referencing the peak latency specifically but more broadly is in agreement with previous interpretations about the effects of training. 

18. Ln 348: “…with a reduced amplitude post-training compared to pre-training” -> But your results show that P2 peak amplitude was enhanced rather than reduced in the post-training task compared to pre-training. Your discussion here does not match what you found. Please revise.

RESPONSE: The full line reads “These results were complemented by the voice changes, which revealed significant changes in response latency and magnitude in that the peak responses occurred earlier and with a reduced amplitude post-training compared to pre-training". This discussion is about the changes in voice amplitude following training and not the ERPs following training. We now put this information related to voice changes in a separate paragraph to help clarify the change to voice changes.

19. Ln 363-266: “Results show that practicing to ignore auditory changes in pitch and hold your voice pitch steady also produced differences in ERP/voice responses, potentially because the non-varying task invoked similar cognitive processes used for voice error detection and correction as the other dynamic or volitional tasks.” -> It seems that the authors try to compare the similarity between pre-training and post-training. However, a more interesting comparison would be whether the instruction during the training session would affect how they respond in the post-training task (i.e., the non-varying task). Please elaborate more on this.

RESPONSE: We agree that this question is of importance and was in fact one of the main motivators of this study (what is the impact of study instructions on post-training performance). However, no time x group effect was found in this study, indicating that all groups had a similar response to the training intervention. We interpret this lack of effect of instruction as outlined in the manuscript that the holding steady condition required the participants to volitionally pay attention to their auditory feedback and that the cognitive processes involved in this act of attending to the feedback which were present in all 3 groups were primarily responsible for the impact of training on the ERP/voice response.

20. Ln 393-394: “in both ERPs and voice responses, the peak responses occurred earlier and with a reduced amplitude post-training compared to pre-training.” -> Your N1 was reduced but N2 was enhanced. So this conclusion should be revised, not “both” ERPs.

RESPONSE: We were not intending to imply that the same result was observed for both N1 and P2, but rather for voice and ERP N1s. The text has been modified to clarify this:

Changes were seen in ERP responses and voice responses, whereby the peak responses occurred earlier and the peak amplitude was modified post-training compared to pre-training. 

Reviewer 2:

1. The motivation and methodological choices for the study require additional information to fully evaluate the contributions. The authors suggest in the introduction and abstract that the current investigation has therapeutic potential, but this notion is not clear from the manuscript in its current form. More information in the motivation and interpretation of the study is required to support this claim throughout. In addition, the specific study aims and hypotheses that correspond to the methods are absent from the introduction. This information is required to fully evaluate the appropriateness of the study methods, analysis, and interpretation of the results. Detailed comments below.

Line 59 – 66. It is not clear why understanding the nature of this response would be useful for voice rehabilitation at this point in the manuscript. A discussion of how individuals seeking voice rehabilitation have differing responses to these paradigms is needed.

RESPONSE: This is not the focus of the paper, but it is a possible avenue. We decided to reword and delete some of this content.

2. Line 75 - 76. Please define what the authors intend by ‘voice control’ and ‘pitch control’. If these are referring to the same thing, it would also be helpful to keep terminology consistent as well.

RESPONSE: Yes, these refer to the same thing. We now use “voice control” throughout.

3. Line 84. Pitch-shift task is not defined. These tasks are variable in the literature, including timing, shift amount, frequency of shifts, duration of shifts, etc. Please include more detail about the paradigms throughout the introduction and review of prior work for clarity.

RESPONSE: We added a definition earlier in the text. We also updated this line to reflect the basic paradigm information about pitch shift timing and amounts presented in the Tumber manuscript. 

4. Line 91. ERP is not defined or explained at this point. The reader needs more information to evaluate what the contribution of this study is to the current study motivation.

RESPONSE: We now introduce the terminology of ERP at this earlier point in the manuscript and clarify that event related potentials to auditory stimuli are measured during the pre and post training conditions. 

5. Line 92. N1 and P2 are not defined or explained at this point.

RESPONSE: We now clarify that the N1 and P2 are auditory evoked potentials that are recorded during electroencephalographic measurement in the study that is described. Note that it is not standard to define N1 and P2 further than to the extent that we provide and would require significantly more text to provide the reader with a basic understanding of the event-related potentials (see our cited literature, for example:

23. Li W, Guo Z, Jones JA, Huang X, Chen X, Liu P, et al. Training of working memory impacts neural processing of vocal pitch regulation. Sci Rep.2015;5:16562.doi: 10.1038/srep16562. 

24. Behroozmand R, Karvelis L, Liu H, Larson CR. Vocalization-induced enhancement of the auditory cortex responsiveness during voice F0 feedback perturbation. Clin Neurophysiol. 2009;120(7): 1303-1312. doi: 10.1016/j.clinph.2009.04.022.

6. Line 95 – 100. The authors motivate the study purpose by saying that no evidence of this yet exists. This could be further strengthened by the addition of why these questions, specifically, need to be answered.

RESPONSE: The importance of identifying this evidence as minimum parameters under which training paradigms operate and its impact on the clinical feasibility of interventions is now identified. 

7. Line 111. This is the first mention of auditory-motor ERPs. Please provide more information here or earlier in the introduction.

RESPONSE: For clarity we now describe these ERPs as auditory-evoked measures. Details about the auditory evoked N1-P2 are introduced earlier (see #5 above). 

8. Line 116. The authors use inconsistent nomenclature for “f0”. Here, it is f0 and elsewhere in the manuscript the ‘f’ is italicized. It would be helpful to use the notation recently proposed as a consensus; a lower-case italic f and subscript of an ‘o’ for oscillation (see Titze, I. R., Baken, R. J., Bozeman, K. W., Granqvist, S., Henrich, N., Herbst, C. T., ... & Wolfe, J. (2015). Toward a consensus on symbolic notation of harmonics, resonances, and formants in vocalization. The Journal of the Acoustical Society of America, 137(5), 3005-3007.)

RESPONSE: We now use consistent nomenclature following the consensus panel throughout the manuscript.

9. Line 116 – 130. It is nicely outlined how the authors are building on the study by Hain and colleagues, however the reason for this additional investigation could be better explained. Specific research aims of the current work are not described. Study hypotheses are also absent. Please include the specific study aims and hypotheses for this work.

RESPONSE: We now clarify the specific aims and hypotheses as follows.

The specific aims of this study were to examine the impact of three brief volitional training paradigms on 1) auditory-motor ERPs (the N1-P2 complex) and 2) voice responses in a pitch-shift task. We predict 1) shorter latencies in the N1 and P2 auditory motor response following volitional training; 2) larger amplitudes in the N1 and P2 auditory motor response following volitional training; and 3) shorter latencies and amplitudes in the voice response during a pitch-shift task following training.

10. Line 147 – 149. The groups are not balanced by sex, which could have impacted the results. Pitch-shift responses have been shown to be impacted by speaker sex (Chen, Z., Liu, P., Jones, J. A., Huang, D., & Liu, H. (2010). Sex-related differences in vocal responses to pitch feedback perturbations during sustained vocalization. The Journal of the Acoustical Society of America, 128(6), EL355-EL360.). This should be acknowledged, along with other study limitations, with references to the specific directional effects that might be expected.

RESPONSE: Limitations of the study including the potential implications of gender effects are now presented in the discussion section.

11. Line 231 – 236. More descriptive statistics are needed for the removed data. E.g., How many trials were removed for each speaker on average, and how many speakers required trials to be removed? A response of greater or less than 10 semitones is very large, how many speakers demonstrated a response outside this threshold? This information informs future work and is important to clarify.

RESPONSE: The outlier removal process occurred within a trial. The process described did not pertain to trial removal. Pitch extraction is known to produce errors in computation. The steps taken were simply to reduce invalid data points due to errors in pitch extraction (see below). The last line indicates trial rejection criteria, however, this rarely occurs in less than 5% of trials:

Then outliers were removed from each trial using several processes including normalization by setting the mean baseline voice pitch to 0 cents and removal of extreme values in the vocalization wave prior to the pitch-shift (for threshold = 30 cents, where max cents > threshold, and min cents < -threshold were rejected) and in the entire duration of each trial (for threshold = 1000 cents, where the whole wave was rejected if max cents > threshold or min cents < -threshold). 

12. Line 236. Why were only responses that opposed the direction of the pitch shift used? This requires justification. Recent work supports that following responses are common, with one study observing that all of their participants had a proportion of following responses to pitch-shifts (Franken, M. K., Acheson, D. J., McQueen, J. M., Hagoort, P., & Eisner, F. (2018). Opposing and following responses in sensorimotor speech control: Why responses go both ways. Psychonomic Bulletin & Review, 25(4), 1458-1467.). It is not clear why following responses were removed from the current sample when evidence shows it is a consistently observed behavior that is relevant to understanding the nature of the pitch-shift response.

RESPONSE: We agree that following responses are consistently observed behavior. The presence of both opposing and following responses in fact motivated this study. However, we were analyzing the compensatory responses here as they present in a majority of trials, at least in the studies conducted in CL’s lab (observationally). An aggregation of following and compensatory responses would simply cancel the compensatory responses as they oppose one another in the direction representation of the compensatory response, which was of primary interest in this study. 

13. Line 236. It would also be interesting to include if there were differing percentages of following responses by group (opposing, following, and non-varying).

RESPONSE: We agree, although it was not feasible to obtain these percentages at the time of writing, due to updates in the analysis software as this was custom-designed MIDI software for data collection. 

14. Line 238 – 240. More information is required on the statistical testing. Please include descriptive information about each statistical test (e.g., method, software used, input variables, outcome variables), any corrections that were applied, and the corresponding study aim/purpose for the test.

RESPONSE: The methods and data analysis section have been reorganized to include and clarify information on the statistical testing (including software used and variables assessed) in the methods section of the manuscript.

15. Line 260. Please include statistical methods and software in the methods section.

RESPONSE: This information was previously provided intext in the results section of the manuscript. All statistical information was reorganized as suggested and moved into the methods and data analyses sections for clarity.

---

## [Editor Report · Decision Letter 1]

2 Oct 2022

PONE-D-22-14364R1Effects of Sensorimotor Voice Training on Event-Related Potentials to Pitch-Shifted Auditory FeedbackPLOS ONE

Dear Dr. Patel,

Thank you for submitting your manuscript to PLOS ONE. I was one of the reviewers in the first-round review process and now take on the role of Guest Academic Editor for this manuscript. After reading your revised manuscript, I feel it is much improved. I have a few minor comments which you can find below in this email. I invite you to submit a revised version of the manuscript that addresses the points raised here. 

We look forward to receiving your revised manuscript.

Kind regards,

Li-Hsin Ning

Guest Editor

PLOS ONE

Journal Requirements:

Additional Editor Comments:

Line 40: It would be better to specify how the training methods influenced the brain and vocal responses in the abstract. Did they become larger or smaller after training?

Line 42: "...in the pre-training..." How did the training methods affect the pre-training task? I think only the post-training task can be affected by the training methods.

Line 91: "...modifications..." Can you specify the change of N1 and P2? Were they reduced or enhanced?

Line 262: You mentioned that up and down responses were aggregated so that the main effect of direction was not tested. However, in Line 337, you reported the main effect of direction (F(1,11) = 5.541, p < 0.05). I wonder which way is correct.

Line 293-294: It would be better to add standard deviations to the mean values (129.32 ms and 148.94 ms). Same for Line 304 and Line 310.

Line 305 and Line 313: There was a significant interaction between electrode and group on P2 latency and P2 amplitude. More elaborations should be made here. What is the implication for this interaction?

Line 320: "...modulated..." Can you specify the change of N1 and P2? Were they reduced or enhanced?

Line 347: "...(cite)." Citations should be added here.

Line 432: "...modified..." reduced or enhanced?

Additional comments:

1. Both reviewers questioned the proportions of following responses in the pre- and post-training tasks. If the information cannot be retrieved, it, at least, should be mentioned or discussed in the limitations.

2. The response to Reviewer 1's 5th comment can be added to the main text: the rationale of using 1000-ms long and 200-ms long stimuli.

---

## [Author Response · Author response to Decision Letter 1]

16 Nov 2022

Response to reviewers document is attached.

---

## [Editor Report · Decision Letter 2]

21 Nov 2022

Effects of Sensorimotor Voice Training on Event-Related Potentials to Pitch-Shifted Auditory Feedback

PONE-D-22-14364R2

Dear Dr. Patel,

We’re pleased to inform you that your manuscript has been judged scientifically suitable for publication and will be formally accepted for publication once it meets all outstanding technical requirements.

Kind regards,

Li-Hsin Ning

Guest Editor

PLOS ONE

Additional Editor Comments (optional):

I appreciate the great efforts that the authors have made in response to my questions and concerns. I have one more suggestion: In Ln 343, the keyword "response latency" should be added to the sentence so that we can be clear that the statistical report refers to the results of response latency, not response amplitude.
---

## [Editor Report · Acceptance letter]

10 Jan 2023

PONE-D-22-14364R2 

Effects of Sensorimotor Voice Training on Event-Related Potentials to Pitch-Shifted Auditory Feedback 

Dear Dr. Patel:

I'm pleased to inform you that your manuscript has been deemed suitable for publication in PLOS ONE. Congratulations! Your manuscript is now with our production department. 

Kind regards, 

on behalf of

Dr. Li-Hsin Ning 

Guest Editor

PLOS ONE